# CAN LLMS EFFECTIVELY LEVERAGE GRAPH STRUCTURAL INFORMATION: WHEN AND WHY

## ABSTRACT

This paper studies Large Language Models (LLMs) augmented with structured data–particularly graphs–a crucial data modality that remains underexplored in the LLM literature. We aim to understand when and why the incorporation of structural information inherent in graph data can improve the prediction performance of LLMs on classifying texts. To address the "when" question, we examine a variety of prompting methods for encoding structural information, in settings where textual node features are either rich or scarce. For the "why" questions, we probe into two potential contributing factors to the LLM performance: data leakage and homophily. Our exploration of these questions reveals that (i) LLMs can benefit from structural information, especially when textual node features are scarce; (ii) there is no substantial evidence indicating that the performance of LLMs is significantly attributed to data leakage; and (iii) the performance of LLMs on a target node is strongly positively related to the local homophily ratio of the node.

## 1 INTRODUCTION

Large Language Models (LLMs) have gained great popularity for a broad range of applications (Brown et al., 2020; OpenAI, 2023). One important reason for their widespread adoption is the ability of an LLM to act as a versatile model, capable of solving a variety of tasks in a zero- or few-shot fashion. Recently, there is an increasing interest in enhancing the versatility of LLMs through multi-modal capabilities (Yin et al., 2023; Yang et al., 2023). Several modalities, including images (Radford et al., 2021), videos (Li et al., 2023), and even robotics (Brohan et al., 2023), have been intensively explored; yet structured data, particularly in the form of graphs, remains largely underexplored. This leads us to an intriguing question: could the incorporation of structural information (such as graphs), when available, improve the predictive accuracy of LLMs?

Directly answering this question turns to be tricky. Consider citation networks as an example, where each node represents a research paper, and each edge indicates a citation relationship between papers. While LLMs can make predictions based on node-level information alone, such as a paper's title and abstract, there has not been a systematic understanding on whether LLMs can benefit from the neighborhood surrounding the target node. A few studies have touched on incorporating structured data with LLMs (Wang et al., 2023; He et al., 2023; Chen et al., 2023). A recent work concurrent to this study, Chen et al. (2023), suggests that LLMs can, in some cases, benefit from neighborhood information, although the extent of this benefit can be dataset-dependent and the underlying mechanisms are not fully understood. Indeed, a notable concern arises as most node classification benchmarks have a data cut-off that predates the training data cut-off of LLMs like ChatGPT. This discrepancy raises concerns about data leakage–LLMs may have seen and memorized at least part of the test data of the common benchmark datasets–which could undermine the reliability of studies using earlier benchmark datasets.

To this end, this paper focuses on two concrete questions relevant to the incorporation of structural information into LLMs. Firstly, we seek to understand the conditions under which incorporating structural information improves the prediction accuracy of LLMs. Secondly, we examine potential factors contributing to the performance of LLMs (either desirable or not), particularly *data leakage* and *homophily* (McPherson et al., 2001), the latter being the tendency of nodes with similar characteristics to connect. As an early attempt towards these questions, we focus on prompting methods for

encoding structural information throughout this study, and leave the investigation of more advanced methods to future work.

Addressing the first question, we examine various methods to encode structural information into prompts, and using ChatGPT API (OpenAI, 2022), we test them on node classification datasets with textual features. Document classification is a very classic language task, and we found it can be naturally augmented with structural context by borrowing popular node classification datasets. In particular, we transform the textual content of a target node and its neighboring nodes into natural language and instruct LLM to make predictions. By varying the richness of node-level textual information and the information incorporated from neighboring nodes, we reveal the conditions under which LLMs would benefit more from structural information.

For the second question, we first investigate the extent to which data leakage might artificially inflate the performance of LLMs. To rigorously measure the data leakage effect, we collect a new dataset, ensuring that the test nodes are sampled from time periods post the data cut-off of ChatGPT. Additionally, we examine the impact of homophily on the classification performance of LLMs. Through controlled experiments and correlation analyses, we establish a relationship between the local homophily ratio and the prediction accuracy of LLMs.

Our key findings are summarized as follows. (i) LLMs benefit more from structural information when textual information of the target node is scarce. (ii) There is no strong evidence that data leakage is a major factor contributing to the performance of LLMs on node classification benchmark datasets. (iii) Homophily in the graph-structured data is a significant contributor to the improved accuracy observed in LLMs after incorporating structural information.

Overall, this study marks an early attempt for the ambitious goal of enabling LLMs to be effectively augmented with structured data, an important data modality. By adapting node classification datasets with textual features, we establish a proper testbed for this goal. We have also examined various prompting methods for encoding the structural information with deeper understandings of their performance.

## 2 RELATED LITERATURE

**LLMs for graph learning.**    We make a distinction between two lines of research: Using LLMs to solve graph learning tasks, and augmenting LLMs with structured data.

The first line has been examined by a few studies recently. He et al. (2023) propose a method where LLMs perform zero-shot predictions along with generating explanations for their decisions, which are then used to enhance node features for training Message Passing Neural Networks (MPNNs) (Gilmer et al., 2017) to predict node categories. Chen et al. (2023) extend the work of He et al. (2023) by using LLMs both as feature enhancers and as predictors for node classification. They offer several observations such as Chain-of-thoughts is not contributing to performance gains. Wang et al. (2023) introduce NLGraph to benchmark LLMs on traditional graph tasks, while Guo et al. (2023) perform an empirical study on using LLMs to solve structure and semantic understanding tasks. More recently, Ye et al. (2023) propose InstructGLM for the instruction tuning of LLMs, like LLaMA (Touvron et al., 2023), for node classification tasks. One commonality for many of these methods is that they use LLMs as a sub-component (e.g., as a feature extractor) of conventional graph learning framework. Our study differs with this line of research in terms of the motivation: while we are using node classification datasets as a testbed, our primary goal is to understand LLMs' capability of processing the graph modality, instead of leveraging LLMs to better solve node classification tasks.

On the other side, the line of research for augmenting LLMs with structured data, which our work belongs to, has also been explored in literature. Works by Zhang (2023) and Jiang et al. (2023) start to explore this space by interfacing LLMs with external tools and enhancing reasoning over structured data like knowledge graphs (KGs) or tables. Pan et al. (2023) further investigate this by outlining a roadmap for integrating LLMs with KGs. However, structured data other than KGs and tables are still underexplored. Despite these initial efforts, a comprehensive understanding of the circumstances under which LLMs can efficiently leverage structural information in a zero-shot setting remains elusive. Our work contributes to this emerging field, seeking to provide more insights into the effective integration of LLMs with structured data.

**Data leakage in LLMs.** Data leakage in LLMs has become a focal point of discussion due to the models' intrinsic ability to memorize training data. As demonstrated by Carlini et al. (2022), LLMs can emit memorized portions of their training data when appropriately prompted, a phenomenon that intensifies with increased model capacity and training data duplication. While memorization is inherent to their function, it raises serious security and privacy concerns. A study by Carlini et al. (2021) shows that extraction attacks can recover sensitive information such as personally identifiable information (PII) from GPT-2 (Radford et al., 2019). This capability to store and potentially leak personal data is further explored by Huang et al. (2022), confirming that although the risk is relatively low, there is a tangible potential for information leakage. Specifically, Carlini et al. (2022) show that the 6 billion parameter GPT-J model (Wang & Komatsuzaki, 2021) memorizes at least 1% of its training dataset. Furthermore, the issue of data leakage complicates the evaluation of these models. As highlighted by Aiyappa et al. (2023), the closed nature and continuous updates of models like ChatGPT make it challenging to prevent data contamination, affecting the reliability of evaluation on LLMs in various applications. In node classification tasks, a concurrent work by Chen et al. (2023) observe that a specific prompt alteration significantly improved performance on `OGBN-ARXIV`, raising concerns about potential test data leakage. In this work, we take a rigorours approach by curating a new dataset for node classification tasks, which is explicitly designed to address the data leakage issues in existing benchmarks.

**Homophily in graph learning.** The concept of homophily (McPherson et al., 2001), which describes the tendency of nodes to form connections with similar nodes, plays an important role in the effectiveness of various graph learning methods (Zhu et al., 2020; Halcrow et al., 2020; Maurya et al., 2021; Lim et al., 2021). The principle of homophily enables MPNNs to smooth node representations by aggregating features from their likely similarly-labeled neighboring nodes. This aggregation process is particularly effective in various types of real-world graphs, such as political networks (Knoke, 1990), and citation networks (Ciotti et al., 2016). Despite its benefits, the reliance on homophily presents a challenge: MPNNs tend to underperform in graphs characterized by heterophily, where connected nodes are likely to differ in properties or labels (Zhu et al., 2020). Notably, the impact of homophily on the integration of structured data into LLMs remains an open area for exploration.

# 3 WHEN AND WHY CAN LLMs BENEFIT FROM STRUCTURAL INFORMATION?

## 3.1 RESEARCH QUESTIONS

In this section, we aim to gain a deeper understanding of two central questions. Firstly, under what circumstances can LLMs benefit from structural information inherent in the data (the "when" question)? Furthermore, what factors can be attributed to LLMs's performance (the "why" question)? To ground our study, we experiment with the ChatGPT API on node classification datasets that have textual node features. We also decompose the questions into hypotheses of finer granularity, as described below.

**The when question.** We hypothesize that the usefulness of structural information for LLMs on a text classification task depends on 1) the prompting methods used to encode the structural information; and 2) the richness of the textual information of each target node. To this end, we explore a variety of prompting methods under two distinct settings, one with *rich textual context* and another with *scarce textual context*. The detailed experimental design and results are discussed in Section 3.2.

**The why question.** Motivated by existing literature in LLM evaluation and graph learning, we hypothesize that *data leakage* and *homophily* are two potential contributing factors to the LLM performance on text classification tasks. While the latter is acceptable and even desirable, the former is not. We investigate the potential impact of data leakage in Section 3.3. In Section 3.4, we examine the role of homophily in the performance of LLMs augmented with structural information.

## 3.2 INFLUENCE OF STRUCTURAL INFORMATION ON LLMs UNDER VARYING TEXTUAL CONTEXTS

We study the impact of structural information on LLM predictions across four node classification benchmark datasets with textual node features: CORA (McCallum et al., 2000; Lu & Getoor, 2003; Sen et al., 2008; Yang et al., 2016), PUBMED (Namata et al., 2012; Yang et al., 2016), OGBN-ARXIV (Hu et al., 2020) and OGBN-PRODUCT (Hu et al., 2020).[1] We create prompts that encode both the textual features and the local graph structure of a target node in natural language, and then request ChatGPT API to make predictions for the target node.[2] The prompt for each node is formulated in one of several styles, as we introduce in details below. Additionally, a fixed dataset-level instruction is attached to the prompt when the prompt is sent to the ChatGPT API. The dataset-level instructions are listed in Table 6 in Appendix A.

**Prompt styles.** Here we introduce the design of prompt styles in our experiments. The exact prompt templates can be found in Table 1.

We first have a few prompt styles that do not encode structural information.

- *Zero-shot*: LLMs make zero-shot predictions based on the target node's textual features only.
- *Few-shot*: LLMs make predictions on nodes' textual features only but with few-shot examples from the training set.
- *Zero-shot Chain-of-Thought (CoT)*: Adding "Let's think step by step" to the end of the zero-shot prompt (Kojima et al., 2022). This simple change has been shown to boost LLMs' performance on various tasks comparable to CoT prompts (Wei et al., 2022).

Then we have two strategies for prompt design conceptually inspired by MPNNs, where information from neighboring nodes is aggregated to enhance the representation of the target node:

The first strategy incorporates randomly selected neighbors into the prompt. The idea behind this strategy is to aggregate information from neighboring nodes, following the paradigms of GCN (Kipf & Welling, 2016) and GraphSAGE (Hamilton et al., 2017). The inclusion of 1-hop neighborhood information in the prompt can be seen as an analogous operation to a single-layer aggregation in GCN, where messages from direct neighbors are aggregated. Specifically, we have two styles:

- *$k$-hop title*: LLMs make predictions based on the target node's textual features as well as titles of neighbors up to k-hop.
- *$k$-hop title+label*: In addition to *$k$-hop title*, we include the labels for neighbors in training set or validation set .

The second strategy is designed to weigh the influence of neighboring nodes during the prediction process. This strategy is inspired by Graph Attention Networks (GAT) (Veličković et al., 2017), which employ attention mechanisms to dynamically allocate weights to neighboring nodes based on their task-specific importance. The strategy consists of two steps. a) *Attention extraction*: the LLM ranks neighbors based on their relevance to the target node. b) *Attention prediction*: the LLM makes predictions based on the target node and top-ranked neighbors. We name the whole strategy as *$k$-hop attention* in our experiment results.

**Richness of textual node features.** To examine how the richness of the textual node features affects text classification accuracy, we compare two different settings:

- *Rich textual context.* In this setting, the nodes are associated with abundant textual features. Specifically, in citation networks (CORA, PUBMED and OGBN-ARXIV), both the paper title and abstract are associated with each node as textual features. In the co-purchasing network (OGBN-PRODUCT), both the product title and product content are associated with each node as textual features. This setting is adopted by several prior studies (Chen et al., 2023; Ye et al., 2023; Guo et al., 2023; Wang et al., 2023; He et al., 2023).

---

[1]Please see Appendix B.1 for the details of the datasets.

[2]We have used `gpt-3.5-turbo-0613` for throughout the experiments.

Table 1: Prompt styles and their corresponding templates. For the style "$k$-hop title+label", we only include the labels for neighbor nodes in training set or validation set. The "attention extraction" and "attention prediction" are respectively the two steps of prompts for the $k$-hop attention strategy.

| Prompt Style | Prompt Template |
| --- | --- |
| Zero-shot | Abstract: \<abstract\>\nTitle: \<title\>\nDo not give any reasoning or logic for your answer. \n\n |
| Zero-shot CoT | Abstract: \<abstract\>\nTitle: \<title\>\nAnswer: \n\nLet's think step by step. \n |
| Few-shot | Abstract: \<few-shot abstract\>\n... \nAnswer: \n\n\<few-shot label\>\n... (more few-shot examples)\nAbstract: \<abstract\>... \nAnswer: \n\n |
| $k$-hop title, $k$-hop title+label | Abstract: \<abstract\>\nTitle: \<title\>\nIt has following neighbor papers at hop 1:\nPaper 1 title: \<paper 1 title\>\nLabel: \<paper 1 label\>\n... (more 1-hop neighbors)\nIt has following neighbor papers at hop 2:\n... (more 2-hop neighbors)\nDo not give any reasoning or logic for your answer. \nAnswer: \n\n |
| Attention extraction | The paper of interest is \<title\>. Please return a Python list of at most \<k\> indices of the most related papers among the following neighbors, ordered from most related to least related. If there are fewer than \<k\> neighbors, just rank the neighbors by relevance. The list should look like this: [1, 2, 3, ...]\n1: \<neighbor title 1\>\n... (more 1-hop neighbors) \n |
| Attention prediction | Abstract: \<abstract\>\nTitle: \<title\>\nIt has following important neighbors, from most related to least related:\n(more neighbors chosen by attention)\nDo not give any reasoning or logic for your answer. \nAnswer: \n\n |

- *Scarce textual context.* In this setting, the nodes are associated with limited textual features. In citation networks (CORA, PUBMED and OGBN-ARXIV), only the paper title is used as textual features. In product networks (OGBN-PRODUCT), only the product name is associated with each node as textual features. While this setting is less explored in the literature, it is of great practical importance due to the prevalence of short texts in social networks (Alsmadi & Gan, 2019). Such limited textual features present challenges like feature sparseness and non-standardization, reducing the effectiveness of traditional methods (Song et al., 2014). In such scenarios, we expect the structural information becomes more useful for the predictions.

**Experimental results.** The experimental results of different prompting methods under the two settings with different richness of textual context are shown in Table 2. We have the following observations:

- Incorporating structural information in prompts brings more gain when textual information about the target node is limited. In rich textual context, zero-shot predictions are very strong baselines because prompts with structural information yield marginal gains on OGBN-ARXIV, PUBMED, and OGBN-PRODUCT (1.6% average increase). This suggests that abundant textual features often suffice for LLMs to make predictions even without structural information. However, in scarce textual contexts, LLMs gain significantly more improvement in accuracy by incorporating structural information compared to rich textual contexts, suggesting that structural information is more important when textual information is limited.

- Few-shot and zero-shot CoT prompts do not yield significant performance gains. Sometimes, they even underperform zero-shot prompts.

- In both rich and scarce textual contexts, the difference of performance between prompting styles that encode structural information ($k$-hop title, $k$-hop title+label and $k$-hop attention) is minimal. This underlines that the availability of textual information is a more critical factor of performance than the specific prompting style used.

Table 2: Classification accuracy for the OGBN-ARXIV, CORA, PUBMED, and OGBN-PRODUCT datasets. ↑ denotes the improvements of best prompt style that leverages structural information over zero-shot method. Best results are **in bold**.

| Textual context | Prompt style | OGBN-ARXIV | CORA | PUBMED | OGBN-PRODUCT |
|---|---|---|---|---|---|
| Rich | Zero-shot | 74.0 | 66.1 | 88.6 | 83.7 |
| | Few-shot | 72.9 | 65.1 | 85.0 | 83.8 |
| | Zero-shot CoT | 71.8 | 56.6 | 81.9 | 80.5 |
| | 1-hop title+label | **75.1** | 72.5 | 89.1 | 85.2 |
| | 2-hop title+label | 74.5 | **74.7** | **89.7** | **86.2** |
| | 1-hop attention | 74.7 | 72.5 | 88.8 | **86.2** |
| | ↑ | 1.1 | 8.6 | 1.1 | 2.5 |
| Scarce | Zero-shot | 69.8 | 61.8 | 85.7 | 78.5 |
| | 1-hop title | 72.3 | 69.6 | 84.8 | 80.5 |
| | 1-hop title+label | **74.3** | 73.9 | 86.4 | 85.3 |
| | 2-hop title | 71.3 | 69.9 | 86.2 | 80.6 |
| | 2-hop title+label | 74.2 | 74.5 | **86.9** | **85.4** |
| | 1-hop attention | 71.3 | **74.7** | 85.1 | 83.9 |
| | ↑ | 4.5 | 12.9 | 1.2 | 6.9 |

In conclusion, structural information offers more benefits for text classification in scarce textual contexts than in rich textual contexts. Next, we further delve into potential factors contributing to the performance of LLMs on text classification tasks.

### 3.3 DATA LEAKAGE AS A POTENTIAL CONTRIBUTOR OF PERFORMANCE

While LLMs have achieved decent performance on the node classification datasets, there is a risk that the performance of LLMs is artificially inflated by data leakage. Note that most node classification benchmark datasets have a data cut-off at 2019 (see Table 7 in Appendix B.1), and ChatGPT was trained on data up to September 2021 (OpenAI, 2023). While the training dataest of ChatGPT is not publicly available, given the widespread of these datasets on the internet and the enormous training corpus of ChatGPT, it is reasonable to worry about the data leakage issue on these datasets.

To this end, we curate a new node classification dataset, ARXIV-2023, which is designed to resemble OGBN-ARXIV as much as possible except that the test nodes are chosen as arXiv Computer Science (CS) papers published in 2023. With the new dataset, we can rigorously investigate the influence of data leakage by comparing the LLM performance between ARXIV-2023 and OGBN-ARXIV.

**Dataset collection.** While, ideally, we should curate the new dataset by simply extending OGBN-ARXIV by including new papers, this is practically challenging for a couple of reasons. In particular, OGBN-ARXIV represents arXiv CS papers in the Microsoft Academic Graph (MAG) until 2019 (Hu et al., 2020), where MAG is a heterogeneous graph representing scholarly communications (Wang et al., 2020). Unfortunately, MAG and its APIs were retired in 2021 and no subsequent data is available.[3] Furthermore, the pipeline to collect and construct MAG is not publicly released. Consequently, we develop our own data collection pipeline to create ARXIV-2023. Specifically, we first sample test nodes from arXiv CS papers published in 2023, and then gather papers within a 2-hop of these test nodes to create a citation network. More details about collection can be found in Appendix B.2.

**Comparison between ARXIV-2023 and OGBN-ARXIV.** As can be seen in Table 3, ARXIV-2023 and OGBN-ARXIV share great similarities in their network characteristics, with consistent in-degree/out-degree pointing to analogous citation behaviors. ARXIV-2023 shows a lower average in-degree in the test set, which is likely because the test papers in ARXIV-2023 are new and have not had much time to accumulate citations. Additionally, Figure 1 illustrates that the label distributions

---

[3] https://www.microsoft.com/en-us/research/project/microsoft-academic-graph/

Table 3: Statistics of OGBN-ARXIV and ARXIV-2023 datasets. Both represent directed citation networks where each node corresponds to a paper published on arXiv and each edge indicates one paper citing another. The metrics In-Degree/Out-Degree, Average Degree, and Published Year are presented for test nodes.

| Dataset | Full Dataset | | Test Set | | |
|---|---|---|---|---|---|
| | #Nodes | #Edges | In-Degree/Out-Degree | Average Degree | Published Year |
| OGBN-ARXIV | 169343 | 1166243 | 1.33/11.1 | 12.43 | 2019 |
| ARXIV-2023 | 33868 | 305672 | 0.16/10.6 | 10.76 | 2023 |

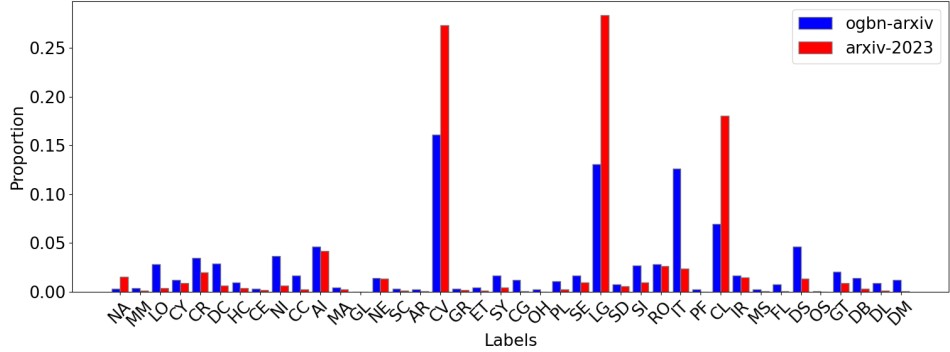

Figure 1: Proportional distribution of labels in OGBN-ARXIV and ARXIV-2023 datasets. Each label represents an arXiv Computer Science Category.

of the two datasets are comparable. A notable trend from ARXIV-2023, in alignment with arXiv statistics,[4] indicates a rise in AI-related categories like ML, LG, CL, reflecting the current academic focus.

Furthermore, we compare the performance of MPNNs on the two datasets. As can be seen from the two bottom rows in Table 4, we observe that the performance metrics for MPNNs (GCN and SAGE) across both datasets are closely matched, suggesting that both datasets present comparable challenges for classification. For a more comprehensive setting of MPNNs, one can refer to Appendix C.

**LLM performance on ARXIV-2023 and OGBN-ARXIV.** If data leakage is a major contributor of performance on OGBN-ARXIV, we would expect the performance drop of LLMs between OGBN-ARXIV (may have leakage problem) and ARXIV-2023 (leakage-free) should be **significantly greater than** the drop on MPNNs on two datasets. This is because LLMs may benefit from their memory on OGBN-ARXIV, but this advantage is not likely on ARXIV-2023. However, as shown in Table 4, we observe **exactly the contrary**: the performance drop of LLMs between OGBN-ARXIV and ARXIV-2023 is less than the drop on MPNNs on two datasets (1.3% compared to 5.1% in rich context, 3.6% compared to 4.5% in scarce context). This means that LLMs actually generalize well to leakage-free data.

To conclude, the observed results neither offer clear evidence in favor of data leakage nor does it advocate that data leakage predominantly improves LLM's performance. Instead, LLM's consistent performance across both datasets stresses its resilience and ability to generalize across varying distribution domains.

### 3.4 IMPACT OF HOMOPHILY ON LLMs CLASSIFICATION ACCURACY

Homophily, the tendency of nodes with similar characteristics to connect, is foundational for many MPNNs. In fact, the degree of homophily in a dataset often correlates with the efficacy of MPNNs

---

[4]https://info.arxiv.org/help/stats/2021_by_area/index.html

Table 4: Comparison between LLM's performance on OGBN-ARXIV and ARXIV-2023. Best results in prompting methods are **in bold**. 1-hop attention means attention extraction and prediction over 1-hop neighbors

| Rich context | | | Scarce context | | |
|---|---|---|---|---|---|
| Prompt style | OGBN-ARXIV | ARXIV-2023 | Prompt style | OGBN-ARXIV | ARXIV-2023 |
| Zero-shot | 74.0 | 73.5 | Zero-shot | 69.8 | 66.6 |
| Few-shot | 72.9 | 73.6 | 1-hop title | 72.3 | **70.7** |
| Zero-shot CoT | 71.8 | 73.7 | 1-hop title+label | **74.3** | 70.4 |
| 1-hop title+label | **75.1** | **73.8** | 2-hop title | 71.3 | 68.9 |
| 2-hop title+label | 74.5 | 73.2 | 2-hop title+label | 74.2 | 68.5 |
| 1-hop attention | 74.7 | 73.7 | 1-hop attention | 71.3 | 69.6 |
| GCN | 75.4 | 70.3 | GCN | 74.8 | 70.3 |
| SAGE | 75.0 | 70.9 | SAGE | 74.4 | 69.1 |

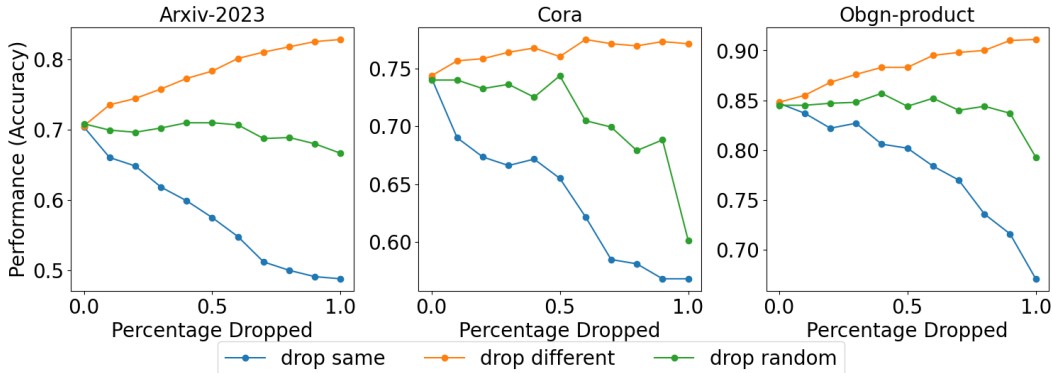

Figure 2: Performance comparison of dropping neighbors using different strategies across ARXIV-2023, CORA, and OGBN-PRODUCT datasets. Three dropping strategies are evaluated: "drop same" removes neighbors with the same label as the target node; "drop different" removes neighbors with different labels as the target node; and "drop random" randomly selects neighbors for removal. When percentage is 1, "drop same" strategy drops all same-label neighbors but preserves all different-label neighbors, and "drop different" strategy drops all different-label neighbors but preserves all same-label neighbors. Details about the strategies are stated in Appendix E.

in classification tasks (Zhu et al., 2020; 2021; Lim et al., 2021; Maurya et al., 2021). Given this significance, it becomes imperative to explore if and how homophily impacts the efficacy of LLMs in similar classification contexts, drawing potential parallels or contrasts with MPNN behaviors.

Since LLM performs node-wise prediction over the neighborhood surrounding the target node, we use *local homophily ratio* (Loveland et al., 2023) to measure the degree of homophily with respect to the target node. For a prompt to predict the category of a target node, the local homophily ratio is defined as the fraction of neighbors sharing the same groundtruth label as the target node over the total number of neighbors included in the prompt. Intuitively, a higher local homophily ratio signals scenarios where a node is surrounded by a greater proportion of neighbors from the same category.

**The neighbor dropping experiment.** We design a controlled experiment to demonstrate the effect of local homophily ratio on prediction accuracy. We gradually drop neighbors in three different ways: a) drop the neighbors with same label as the target node; b) drop the neighbors with different label as the target node; and c) drop neighbors randomly. We include details about the neighbor dropping strategies in Appendix E. The experimental results are shown in Figure 2, where we observe an evident trend: as we selectively remove neighbors sharing the same labels, there's a decrease in prediction accuracy. Conversely, discarding neighbors with different labels leads to an increase in accuracy. This selective dropping inherently modifies the local homophily ratio within the prompts.

Table 5: Point biserial correlation between local homophily ratio and prediction correctness across five datasets (p-values in brackets). Point biserial correlation ranges between $[-1, 1]$, where a value of 1 indicates a perfect positive relationship. A higher correlation value indicates that the local homophily ratio and prediction correctness are more positively related.

| Prompt Style | OGBN-ARXIV | CORA | PUBMED | ARXIV-2023 | OGBN-PRODUCT |
|---|---|---|---|---|---|
| Zero-shot | 0.440 (0.000) | 0.070 (0.106) | 0.278 (0.000) | 0.367 (0.000) | 0.387 (0.000) |
| 1-hop title+label | 0.518 (0.000) | 0.222 (0.000) | 0.443 (0.000) | 0.481 (0.000) | 0.560 (0.000) |

The results show that accuracy of predictions made by LLMs is positively related to local homophily ratio.

**Correlation study.** Building on the insights from the dropping neighbors experiment, we further investigate the relationship between local homophily ratio and the prediction correctness across different datasets. Each node possesses two key attributes: a) its local homophily ratio, which is a continuous random variable in $[0, 1]$, and b) its prediction correctness, which is a binary random variable (0 indicating an incorrect prediction and 1 indicating a correct prediction). To quantify the correlation between these two attributes, we employ the point biserial correlation method (Kornbrot, 2014). This correlation coefficient ranges between -1 and 1, where a value of 1 signifies a perfect positive relationship. The results of our analysis across five datasets are detailed in Table 5.

For the CORA dataset, we observe no significant correlation when only the title is used in prompts. However, a positive correlation emerges when neighbors are included alongside the title. This suggests that the more homophily is incorporated into the prompt, the more accurate the prediction becomes.

For the other datasets, a positive correlation is evident in both the zero-shot and 1-hop title+label settings. In Table 5, zero-shot prediction (the one that doesn't use structural information at all) also showed high correlation with the homophily ratio of the node. This suggests a complicated mechanism for LLMs to perform better on homophilous nodes: those nodes are easier to be classified in the first place; the added structural information has some further contributions.

In summary, our findings underline the critical role of homophily in influencing LLM's text classification performance. The experiments and analyses consistently point to a positive relationship between local homophily ratio and prediction correctness, emphasizing the importance of understanding network structures and node relationships in enhancing classification outcomes.

## 4 CONCLUSIONS AND FUTURE WORK

This study marks an early step towards a broader research aim: enabling LLMs to process structured data, a crucial data modality commonly seen in practice. In this study, we have adapted node classfication datasets with textual features from graph learning benchmarks to establish a testbed for LLMs augmented with structured data. Our preliminary examination on prompting methods for encoding the structural information shows that LLMs benefit more from structural information when the textual features of the target node is scarce. We also delve into the impact of data leakage and homophily, which provides deeper insights about the LLM performance when augmented with graph-structured data.

This study also opens several avenues for future research. Firstly, the findings of this study, as well as the new dataset curated by this work, establish a proper benchmark setup for more advanced methods to encode structural information for LLMs, such as finetuning or adapter training. Secondly, while we find that data leakage is not a major concern for the prompting methods examined in this paper, it is still possible that more advanced methods can elicit the memory of the LLMs from training corpus. We may need further investigation on the data leakage issue when proceeding with evaluating other methods. Finally, the fact that homophily plays a crucial role in the performance gain of LLMs with structured data suggests that LLMs may be utilizing superficial correlational information to aid the prediction tasks. It would be interesting to further investigate whether we can make LLMs grasp the deeper relational structure of the graph data.

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

## A  DETAILS ABOUT PROMPTING FORMAT AND SETTINGS

Our API call to ChatGPT utilize a two-part prompt structure, in line with the ChatGPT Chat Completions API.[5] Each API call involves a system prompt and a user prompt. The system prompt, detailed in Table 6, sets ChatGPT's objective and return format. The user prompt, outlined in Table 1, provides information on the target node and its neighborhood for prediction. To standardize ChatGPT's output format, we append "Do not give any reasoning or logic for your answer" to the end of all prompts, except zero-shot CoT prompts.

Table 6: System prompts for each dataset.

| Dataset | System Prompt |
| --- | --- |
| OGBN-ARXIV, ARXIV-2023 | Please predict the most appropriate arXiv Computer Science (CS) sub-category for the paper. The predicted sub-category should be in the format 'cs.XX'. |
| CORA | Please predict the most appropriate category for the paper. Choose from the following categories:\nRule Learning\nNeural Networks\nCase Based\nGenetic Algorithms\nTheory\nReinforcement Learning\nProbabilistic Methods\n |
| PUBMED | Please predict the most likely type of the paper. Your answer should be chosen from:\nType 1 diabetes\nType 2 diabetes\nExperimentally induced diabetes.\n |
| OGBN-PRODUCT | Please predict the most likely category of this product from Amazon. Your answer should be chosen from the list:\nHome & Kitchen\nHealth & Personal Care\n... |

We outline the settings for each prompting method as follows:

1. *Few-shot*: Two correct example predictions from ChatGPT are added before the target node information.

2. *Target node with neighbors*: For datasets OGBN-ARXIV, CORA, PUBMED and ARXIV-2023, prompts include up to 20 one-hop and 5 two-hop neighbors. For OGBN-PRODUCT, up to 40 one-hop and 10 two-hop neighbors are included.

3. *Attention extraction*: The maximum number of neighbors is the same as *Target node with neighbors*. We only consider one-hop attention in this study, setting the attention number $k$ to 5.

Common settings for all methods include a temperature of 0 and a maximum output token limit of 500. If a neighbor belongs to the training or validation set, its label is included in the prompt.

## B  DATASETS INFORMATION

In this section we detail the information about benchmark datasets and the collection pipeline of ARXIV-2023.

### B.1  DATASETS STATISTICS AND SPLITS

Table 7 presents basic statistics for each dataset. For detailed information on datasets and methods to obtain raw text attributes, please see Appendix A in Chen et al. (2023).

The dataset splits are as follows:

1. CORA: Training/Validation/Testing ratios are 0.1/0.2/0.2.

---

[5]https://platform.openai.com/docs/guides/gpt/chat-completions-api

Table 7: Statistics of datasets. Data cut-off indicates the latest data coverage of the dataset.

| Dataset | #Nodes | #Edges | #Task | Metric | #Test Nodes | Data Cut-Off |
|---|---|---|---|---|---|---|
| CORA | 2,708 | 5,429 | 7 | Accuracy | 542 | 2000 |
| PUBMED | 19,717 | 44,338 | 3 | Accuracy | 1,000 | 2000 |
| OGBN-ARXIV | 169,343 | 1,166,243 | 40 | Accuracy | 1,000 | 2019 |
| OGBN-PRODUCT | 2,449,029 | 61,859,140 | 1 | Accuracy | 1,000 | 2019 |
| ARXIV-2023 | 33,868 | 305,672 | 40 | Accuracy | 668 | 2023 |

2. PUBMED: Training/Validation/Testing ratios are 0.6/0.2/0.2, following He et al. (2023).

3. OGBN-ARXIV: Original OGB (Hu et al., 2020) splits are used, categorizing papers by their publication year: training (pre-2017), validation (2018), and testing (2019).

4. OGBN-PRODUCT: Original OGB splits are used based on sales ranking: top 8% for training, next 2% for validation, and the remainder for testing.

5. ARXIV-2023: Year-based splits similar to OGBN-ARXIVis adopted: training (pre-2019), validation (2020), and testing (2023).

Due to API cost and rate limits, we test on a random sample of 1,000 nodes for PUBMED, OGBN-ARXIV, and OGBN-PRODUCT, using a fixed seed for reproducibility.

## B.2 COLLECTION OF ARXIV-2023

The detailed pipeline is as follows:

1. Sample 668 test nodes from around 46,000 arXiv CS papers published from January 1 to August 22, 2023.

2. Extract references to identify one-hop and two-hop neighbors. References were obtained by two steps. First, we search for valid arXiv IDs within each paper, following a method similar to (Clement et al., 2019). Second, we use AnyStyle to extract the titles of the references,[6] which we then search for via the arXiv API.[7] Titles found on arXiv are considered valid citations if they have a small levenshtein distance (Miller et al., 2009) from the searched title. To prevent duplicate searches, we skip any references that already have a matched arXiv ID. To comply with the arXiv API's rate limit, each paper is restricted to a maximum of 30 searches. For papers published before 2019, we attempt to match them to nodes in the OGBN-ARXIV based on titles. Unmatched pre-2019 nodes are excluded from our dataset.

3. Construct a citation network using nodes from step 2. Basically for each node we need a list of paper it cites. While references for test nodes and one-hop nodes are obtained through both arXiv ID matching and title searching, the references for two-hop nodes are solely determined by arXiv ID matching, due to rate limit constraints. Dataset statistics are in Table 3. We have similar test node degrees between OGBN-ARXIV and ARXIV-2023.

## C   MPNNS AS BASELINES

**Embedding generation.**   We adapt the embedding generation pipeline from Hu et al. (2020) to train a skip-gram model (Mikolov et al., 2013) on corpus comprising titles and abstracts from both OGBN-ARXIV and ARXIV-2023. Each paper's 128-dimensional feature vector is then obtained by averaging the word embeddings in its title.

**Hyperparameter tunning.**   Baseline models GCN and SAGE are implemented with PyG (Fey & Lenssen, 2019). For hyperparameter tunning, we perform a random search on the following hyperparameter tuning range for every model following Ma et al. (2022):

---

[6]https://github.com/inukshuk/anystyle
[7]https://info.arxiv.org/help/api/basics.html

Table 8: Classification accuracy for the OGBN-ARXIV, CORA, ARXIV-2023, PUBMED, and OGBN-PRODUCT datasets on LLaMA-2-7B-chat. ↑ denotes the improvements of best prompt style that leverages structural information over zero-shot method. Best results are **in bold**.

| Textual Context | Prompt Style | OGBN-ARXIV | CORA | ARXIV-2023 | PUBMED | OGBN-PRODUCT |
|---|---|---|---|---|---|---|
| Scarce | Zero-shot | 38.8 | 24.5 | 38.2 | 70.1 | 51.7 |
| | 1-hop title | 51.5 | 44.8 | 45.5 | 70.9 | 52.8 |
| | 1-hop title+label | **58.0** | **71.0** | **53.4** | **75.5** | **78.9** |
| | ↑ | 19.2 | 46.5 | 15.2 | 5.4 | 27.2 |
| Rich | Zero-shot | 45.1 | 18.1 | 45.1 | 71.6 | 51.3 |
| | 1-hop title | 51.6 | 51.5 | 50.0 | 68.8 | 52.1 |
| | 1-hop title+label | **66.9** | **66.7** | **60.2** | **73.0** | **77.2** |
| | ↑ | 21.8 | 48.6 | 15.1 | 1.4 | 25.9 |

- Hidden size: $\{32, 64\}$.

- Learning rate: $\{.001, .005, .01, .1\}$.

- Dropout rate: $\{.2, .4, .6, .8\}$.

- Weight decay: $\{.0001, .001, .01, .1\}$.

Each model is run on 100 random configurations and each random configuration is run for 3 times on OGBN-ARXIV and ARXIV-2023. The max training epoch number is 2000. When training is finished, we use the model with highest average validation accuracy on the dataset for testing.

# D ADDITIONAL ANALYSIS ON THE INFLUENCE OF STRUCTURAL INFORMATION ON LLMS.

## D.1 CLASSIFICATION ACCURACY ON LLaMA-2-7B-CHAT

The results in the main paper are based on `gpt-3.5-turbo-0613`. Here we test the performance of `LLaMA-2-7B-chat`. The results are shown in Table 8. The model gains significant improvement after incorporating structural information in both rich and scarce textual context. The results align with our observation in the paper with ChatGPT that incorporating structural information actually brings performance improvement in both rich and scarce contexts. But a different observation is that the improvement in scarce textual context is not necessarily higher than the improvement in rich textual context. This may be because LLaMA-2 is not able to sufficiently leverage the entire text for the prediction in zero-shot prediction. Combining the results of ChatGPT, the conclusion is that, with powerful enough LLM and rich text (e.g. ChatGPT with rich context), the structural information is marginal. But when the text information is scarce or if the LLM cannot fully utilize the text information, structural information can be significantly helpful.

## D.2 THE NUANCES OF WHEN STRUCTURAL INFORMATION SATURATES ON LLMS AND MPNNS.

We compare the performance increase from incorporating structural information for LLMs and MPNNs respectively in Table 9. The average increase from structural data of ChatGPT on 4 datasets is 2.78% (rich context) and 5.44% (scarce context). But the increase from structural data of MPNNs is 6.98% (rich context) and 14.07% (scarce context), which is significantly higher than the gain of LLMs. It means that The benefit of structural information saturates earlier on ChatGPT than MPNNs.
While it's true that structural information is generally more helpful when text is scarce, **quantitatively ChatGPT behaves differently from GNNs**: the benefit of structural information saturates much earlier than GNNs with moderate rich textual features; and this is non-trivial since LLaMA-2 doesn't saturate as early as ChatGPT. The average increase from structural data on 4 datasets for ChatGPT/MPNNs/LLaMA-2-7B-chat are 2.78%/6.98%/21.7% respectively.

Table 9: Classification accuracy for the OGBN-ARXIV, CORA, ARXIV-2023, PUBMED on ChatGPT as well as GCN, SAGE and MLP. ↑ (LLMs) denotes the improvements of best prompt style that leverages structural information over zero-shot method. ↑ (MPNNs) denotes the improvements of the best MPNNs over MLP (without structural information).

| Textual Context | Prompt Style | OGBN-ARXIV | CORA | ARXIV-2023 | PUBMED |
|---|---|---|---|---|---|
| Rich | Zero-shot | 74.0 | 66.1 | 73.5 | 88.6 |
| | 1-hop title+label | 75.1 | 72.5 | 73.8 | 89.1 |
| | 2-hop title+label | 74.5 | 74.7 | 73.2 | 89.7 |
| | 1-hop title+label, attention | 74.7 | 72.5 | 73.7 | 88.8 |
| | ↑ (LLMs) | 1.1 | 8.6 | 0.3 | 1.1 |
| | MLP | 69.9 | 65.4 | 69.7 | 86.2 |
| | GCN | 75.4 | 83.0 | 70.3 | 88.4 |
| | SAGE | 75.0 | 83.2 | 70.9 | 90.0 |
| | ↑ (MPNNs) | 5.5 | 17.8 | 1.3 | 3.8 |
| Scarce | Zero-shot | 69.8 | 61.8 | 66.6 | 85.9 |
| | 1-hop title | 72.3 | 69.6 | 70.7 | 80.8 |
| | 1-hop title+label | 74.3 | 73.9 | 70.4 | 84.7 |
| | 2-hop title | 71.3 | 69.9 | 68.9 | 83.5 |
| | 2-hop title+label | 74.2 | 74.5 | 68.5 | 86.4 |
| | ↑ (LLMs) | 4.5 | 12.7 | 4.1 | 0.5 |
| | MLP | 61.9 | 55.7 | 58.5 | 82.0 |
| | GCN | 74.8 | 81.2 | 70.3 | 87.1 |
| | SAGE | 74.4 | 78.8 | 69.1 | 87.9 |
| | ↑ (MPNNs) | 13.0 | 25.6 | 11.8 | 6.0 |

# E ADDITIONAL ANALYSIS FOR DATA LEAKAGE

**Details about dropping experiments.** We have three different strategies: a) drop the neighbors with same label (*drop same*), b) drop the neighbors with different label (*drop different*), c) drop neighbors randomly (*drop random*). Let's define $x$ as the number of neighbors with the same ground truth label as the target node, and $y$ as the number of neighbors with a different label from the target node. Given a dropping percentage $p$, we elaborate on the three strategies:

1. *drop random*: We randomly drop $(x + y)p$ neighbors.

2. *drop same*: We retain $\max(x - (x + y)p, 0)$ neighbors with the same labels as the target node while preserving all $y$ neighbors with different labels.

3. *drop different*: We retain $\max(y - (x + y)p, 0)$ neighbors with the different labels from the target node while preserving all $x$ neighbors with same labels.

We further explain this by an example. Assume node $A$ has 10 neighbors and 6 of the neighbors have same labels as $A$. When dropping percentage is 0.5, *drop same* strategy drops 5 nodes with same label, resulting in 1 neighbor with same label and 4 neighbors with different labels. *drop different* strategy drops all 4 nodes with different labels, resulting in 6 neighbors with same label.

**Ablation study on the effect of labels in the prompt** We investigate the possibility that LLMs are relying on a simple majority vote in its prediction. We propose a new neighbor dropping experiment with three different prompting styles for neighbors: (i) 1-hop title+label, (ii) 1-hop title and (iii) 1-hop label. 1-hop label means that we only include the label of the neighboring papers, which is used as an ablation study to gauge whether LLM is performing a majority vote based on label information.

If LLMs do rely on a majority vote to determine its prediction. We would expect that the "drop different" curve with 1-hop label goes higher than 1-hop title+label because we are dropping more and more neighbors with different labels. However, we are not observing this in Figure 3 and 4, and the 1-hop label curve is lower than 1-hop title+label curve. This observation refutes the hypothesis

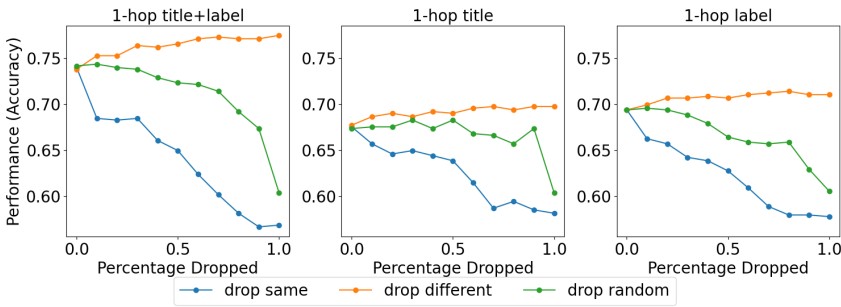

Figure 3: Performance comparison of dropping neighbors using different strategies on CORA dataset. Three dropping strategies are evaluated: (i) 1-hop title+label, (ii) 1-hop title and (iii) 1-hop label

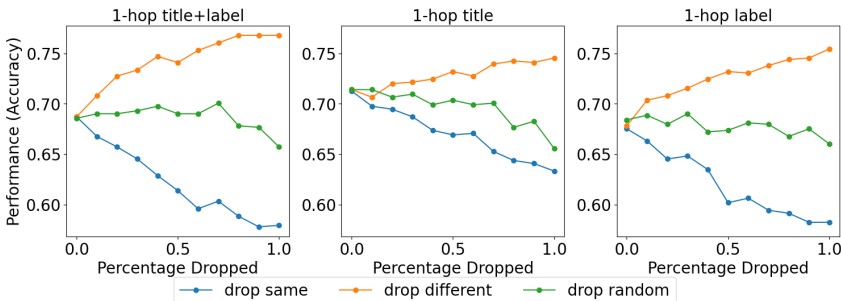

Figure 4: Performance comparison of dropping neighbors using different strategies on ARXIV-2023 dataset. Three dropping strategies are evaluated: (i) 1-hop title+label, (ii) 1-hop title and (iii) 1-hop label

that LLMs rely on simple majority vote for prediction. Instead, including more context information will help LLMs to make more accurate predictions as 1-hop title+label "drop different" curve is higher than 1-hop label "drop different" curve.

**Investigating data leakage through prompt variability.** Chen et al. (2023) reveal considerable fluctuations in Language Model (LLM) performance on OGBN-ARXIV when using three distinct prompt words: "arXiv cs subcategory," "arXiv identifier," and natural language. These variations have been interpreted as potential indicators of data leakage.

To delve deeper into this issue, we expand upon their experiments by testing additional prompt words. We also introduce two experimental settings: one with label options provided and another without. As displayed in Table 10, the relative efficacy of various prompts on OGBN-ARXIV mirrors their performance on ARXIV-2023. Importantly, prompts with options underperform on both datasets, underscoring a consistent trend.

Also, utilizing structural information in the prompts can somewhat mitigate the performance drop from less effective prompts. Indicate that LLMs can leverage structural information to improve predictions. This further supports that there is no conclusive evidence for data leakage.

Table 10: Performance across different prompt types between OGBN-ARXIV and ARXIV-2023.

| System Prompt | Zero-shot | | 1-hop title+label | |
|---|---|---|---|---|
| | OGBN-ARXIV | ARXIV-2023 | OGBN-ARXIV | ARXIV-2023 |
| Please predict the most appropriate arXiv Computer Science (CS) sub-category for the paper. The predicted sub-category should be in the format 'cs.XX'. | 74.0 | 73.7 | 74.3 | 70.4 |
| Please predict the most appropriate arXiv Computer Science (CS) sub-category for the paper. Your answer should be chosen from cs.AI, ..cs.SY. The predicted sub-category should be in the format 'cs.XX'. | 66.0 | 68.1 | 70.7 | 67.9 |
| Please predict the most appropriate original arXiv identifier for the paper. The predicted arxiv identifier should be in the format 'arxiv cs.xx'. | 71.3 | 70.8 | 73.7 | 67.5 |
| Please predict the most appropriate original arXiv identifier for the paper. Your answer should be chosen from cs.ai,.. cs.sy. The predicted arxiv identifier should be in the format 'arxiv cs.xx'. | 58.4 | 57.2 | 71.7 | 64.2 |
| Please predict the most appropriate category for the paper. Your answer should be chosen from "Artificial Intelligence",.. "Systems and Control". | 54.6 | 53.4 | 74.1 | 67.8 |

