# OpenReview forum: "Can LLMs Effectively Leverage Graph Structural Information: When and Why"
_ICLR.cc/2024/Conference — ICLR 2024 Conference Withdrawn Submission_

### Official Review · Reviewer_vRFw · 2023-10-15

**Soundness:** 2 fair
**Presentation:** 3 good
**Contribution:** 2 fair
**Rating:** 3
**Confidence:** 5

**Summary:**

This paper offers valuable insights into the utilization of graph structural information by large language models (LLMs), focusing on the context of node classification on text attributed graphs. In addressing the "when" aspect, the authors investigate the impact of different prompt designs and the richness of textual data. Regarding the "why" aspect, the study delves into the effects of label leakage and homogeneity.

**Strengths:**

- Significance: The paper's exploration of the "when" and "why" behind LLMs' effectiveness in graph-related tasks holds immense significance. Given the increasing prominence of Graph+LLMs in various applications, the insights provided by this work are timely and valuable.

- Quality. The authors dissect the problem into two crucial aspects: "when" and "why." They provide a clear and well-reasoned argument, accompanied by rigorous experimental and analytical support, such as the introduction of the new dataset arxiv-2023.

- Clarity. The paper is  well-structured and clear in its presentation

**Weaknesses:**

- **Overlap with Previous Work.** Several experiments and observations made in the paper have already been addressed in previous or concurrent works, such as [1], [2], and [3]. These prior works have explored topics like the effect of different prompt designs and settings, including zero-shot, few-shot, with k-hop neighbor information, and with title or title and abstract. It is acknowledged that the paper does introduce some unique elements, such as prompt designs with attention analogous to Graph Attention Networks (GAT) and providing more in-depth analysis. However, the experiments appear to have significant overlaps with existing research, leading to a perception of a relatively trivial contribution.
- **Spurious Analysis.**
- Q1: "If data leakage is a major contributor of performance on OGBN-ARXIV, we would expect prompting methods based on LLMs to perform worse than MPNNs on ARXIV-2023."
This argument regarding data leakage and the expected performance of LLMs compared to MPNNs on a new dataset is not convincingly made. We would expect LLMs to perform worse than MPNNs on ARXIV-2023 because there might be no causality between MPNN doing good/bad and LLM also doing good/bad.
- Q2: The statement, "our findings underline the critical role of homophily in influencing LLM’s node classification performance," is made, but it may oversimplify the issue.  While homophily's influence is acknowledged, it's essential to delve deeper into why LLMs are affected by homophily. It is suggested that LLMs tend to employ a simple majority vote approach when provided with neighbor information, regardless of the design of prompts, due to not being directly trained on structured (graph) data.

Reference:

[1] Harnessing Explanations: LLM-to-LM Interpreter for Enhanced Text-Attributed Graph Representation Learning (Table 8) [arxiv link](https://arxiv.org/abs/2305.19523)

[2] Exploring the Potential of Large Language Models (LLMs) in Learning on Graphs (Table 10, Table 16) [arxiv link](https://arxiv.org/abs/2307.03393)

[3] GraphText: Graph Reasoning in Text Space (Table 1) [arxiv link](https://arxiv.org/abs/2310.01089)

**Questions:**

For Q1 and Q2, see weakness above.

Q3: In the new arXiv-2023 datasets, you mention, "Specifically, we first sample test nodes from arXiv CS papers published in 2023, and then gather papers within a 2-hop of these test nodes to create a citation network." Could you please provide the exact number of nodes that belong to the year 2023 in this dataset?

---

> ### Author Response · Authors · 2023-11-18
> **Response to Reviewer vRFw**
>
> Thank you for your thorough review and insightful feedback on our paper. We are grateful for your recognition of the paper's significance on timely topics on “increasing prominence of Graph+LLMs in various applications”, quality of “rigorous experimental and analytical support”, and clarity.
>
> We address your concerns in detail below.
>
>
> > “Overlap with Previous Work. Several experiments and observations made in the paper have already been addressed in previous or concurrent works, such as [1], [2], and [3]. These prior works have explored topics like the effect of different prompt designs and settings, including zero-shot, few-shot, with k-hop neighbor information, and with title or title and abstract. It is acknowledged that the paper does introduce some unique elements, such as prompt designs with attention analogous to Graph Attention Networks (GAT) and providing more in-depth analysis. However, the experiments appear to have significant overlaps with existing research, leading to a perception of a relatively trivial contribution.”
>
> While we acknowledge these papers, **developing new prompts is not our main focus**. For example, one of the major questions that we want to answer here is **how does the richness of target nodes’ textual features affect LLMs’ benefit from structural information**. And the take-away is that LLMs can benefit more from structural information when textual node features are scarce. In our experiment, we compared two settings: Rich textual context (e.g. paper title and abstract are given) and Scarce textual context (only title is given), to show this (Page 6, Table 2).
>
> **The above question is not overlapping with previous works mentioned.**
> 1. The prompts designed in [1] always use title and abstract (Table 7,8), lacking comparison in terms of node-level feature richness.
> 2. [2] also regards the textual features of the target node as a whole, without any control over the richness of the features.
> 3. Although [3] includes a comparison between “feature+label” and “label” in Table. 1, there is no control over the **richness** of the feature. Moreover, [3] is published on arXiv on 10/02/23, which is later than the submission deadline of ICLR 2024 (09/28/23).
>
> > “Q1: "If data leakage is a major contributor of performance on OGBN-ARXIV, we would expect prompting methods based on LLMs to perform worse than MPNNs on ARXIV-2023." This argument regarding data leakage and the expected performance of LLMs compared to MPNNs on a new dataset is not convincingly made. We would expect LLMs to perform worse than MPNNs on ARXIV-2023 because there might be no causality between MPNN doing good/bad and LLM also doing good/bad.”
>
> Sorry for the confusion in L 255, we are not assuming there is a strict relationship between MPNNs performance and LLMs performance. Our logic to support the argument that “there is no substantial evidence indicating that the performance of LLMs is significantly attributed to data leakage” is elaborated in the common response to all authors.
>
> > “Q2: The statement, "our findings underline the critical role of homophily in influencing LLM’s node classification performance," is made, but it may oversimplify the issue. While homophily's influence is acknowledged, it's essential to delve deeper into why LLMs are affected by homophily. It is suggested that LLMs tend to employ a simple majority vote approach when provided with neighbor information, regardless of the design of prompts, due to not being directly trained on structured (graph) data.”
>
> We acknowledge your suggestion to delve deeper into why homophily affects LLMs' performance in node classification. To address the concern that “LLMs tend to employ a simple majority vote approach when provided with neighbor information”, we develop a new neighbor dropping experiment with three new setting with different prompt for neighbors: i) 1-hop title+label, (ii) 1-hop title and (iii) 1-hop label. “1-hop label” means that we only include the label of the neighboring papers. This special design is to **gauge whether LLM is performing a majority vote with extra neighborhood information**. **Please refer to Page 17, Figure. 7,8 in the update paper.**
>
> If LLMs do rely on a majority vote to determine its prediction. We would expect that the “drop different” curve with 1-hop label goes up to 100% accuracy. However, we are not observing this, and the 1-hop label curve is lower than 1-hop title+label curve. **This observation refutes the hypothesis that LLMs rely on simple majority vote for prediction**. Instead, including more context information will help LLMs to make more accurate predictions as 1-hop title+label curve is higher than 1-hop label curve on both cora and arxiv-2023 datasets.
> Reference:
>
> [1] Harnessing Explanations: LLM-to-LM Interpreter for Enhanced Text-Attributed Graph Representation Learning
>
> [2] Exploring the Potential of Large Language Models (LLMs) in Learning on Graphs
>
> [3] GraphText: Graph Reasoning in Text Space

---

> > ### Comment · Reviewer_vRFw · 2023-11-21
> >
> > I would like to thank the authors for the detailed response and the added experiments.
> >
> > **Overlap with Previous Work.** I fully understand that developing new prompts is not your main focus, and I concur with the key takeaway message the authors aim to convey regarding the relationship between the richness of semantics of the target node and the performance of LLM. However, concerning the paper itself, regarding the "when" aspect, the takeaway message is that:
> > - [Semantic Richness] LLMs can benefit from structural information when textual node features are scarce.
> >
> > Regarding the "why" aspect, the message is that:
> > - [Homophily] the performance of LLMs on a target node is strongly positively related to the local homophily ratio of the node. As mentioned by Reviewer Ayoh and Reviewer 2d38, the effect of homophily is well-established and not surprising.
> >
> > - [Label Leakage] there is no substantial evidence indicating that the performance of LLMs is significantly attributed to data leakage; While the new dataset and experiments can rule out potential reasons why LLM works well, they do not provide a definitive answer to the "why" question.
> >
> > In summary, regarding [Semantic Richness], I agree that previous work might not have delved into it. Concerning [Homophily], the message seems trivial and not surprising. Regarding [Label Leakage], it does not essentially answer the "why" question.
> >
> >
> >
> > **Q1.** Thanks for the clarification for Q1.
> >
> > **Q2.** Thanks for the added experiment for Q2. I have a quick question, how did you deal with the nodes with no neighbours after dropping neighbours? Did you include them or filter out them? Because if they are still included, it can justify why the accuracy is not 100% for the 'drop different' curve with 1-hop label
> >
> >
> > Thanks for the authors' efforts in addressing my concern. But I still tend to maintain my score, mainly due to concerns about the overall contribution, i.e., the limited message conveyed as a benchmark/survey/investigation paper.

---

> > > ### Author Response · Authors · 2023-11-21
> > > **Response to Reviewer vRFw**
> > >
> > > Thank you for your insightful feedback. We'd like to emphasize one point regarding homophily.
> > >
> > > The story of homophily is actually quite nuanced: in Table 5, zero-shot prediction (the one that doesn’t use structural information at all) also showed high correlation with the homophily ratio of the node. This suggests a **complicated mechanism for LLMs to perform better on homophilous nodes**: those nodes are easier to be classified in the first place; the added structural information has some further contributions.
> > >
> > > > Q2. Thanks for the added experiment for Q2. I have a quick question, how did you deal with the nodes with no neighbours after dropping neighbours? Did you include them or filter out them? Because if they are still included, it can justify why the accuracy is not 100% for the 'drop different' curve with 1-hop label
> > >
> > > Thank you for pointing out this gap. We indeed kept the nodes with no neighbors left. However even if we keep these nodes, it cannot explain why “1-hop label” is not higher than “1-hop title+label” on cora and ogbn-arxiv, if LLMs are actually making a majority vote. So this experiment still fulfills its purpose of disproving that LLMs are doing a majority vote. We updated our paper to make the statement more precise. (L 593)

---

### Official Review · Reviewer_Ayoh · 2023-10-27

**Soundness:** 2 fair
**Presentation:** 2 fair
**Contribution:** 1 poor
**Rating:** 3
**Confidence:** 4

**Summary:**

The paper presents an experimental study on leveraging structured information (specifically graphs) in LLMs and analyzed the impact of structured information on the performance of LLM, specifically on the node classification task.  The general idea is to augment the textual features of a given node in the prompt with the structured information from the graph by describing the list of k-hop neighbors (grouped by hop levels) along with their textual features. The paper conducts their experimental study using ChatGPT 3.5 turbo on  our popular public graph-based datasets (OGBN-ARXIV, CORA, PUBMED, and OGBN-PRODUCT). Based on their experimental observations, following major conclusions were drawn from the paper: i) LLMs can potentially benefit from structural information, especially when textual features on the individual nodes  is scarce, ii) LLMs performance improves further on graphs that have higher local homphily ratio, i.e. graphs where nodes with similar features are more likely to be connected by an edge, and iii) No conclusive evidence was observed to attribute improvement of performance in LLMs to data leakage.

**Strengths:**

1. The paper presents a  detailed experimental study on the impact of structural information on LLM's performance on node classification tasks while exploring several aspects of the problem, including: i) the impact of local homophily ratio on LLM performance, ii) altering textual information on the node, iii) data-leakage impact (i.e. possibility of test dataset being used in LLM training )
2. The insights seem to be logical and intuitive and I see fair experimental setup being used to backup the claims.

**Weaknesses:**

1. Lack of significant contributions. My biggest concern about this paper is that the conclusions derived from the paper are quite trivial and unsurprising. For instance, the observation that stronger improvements in the performance on data with scarce textual features have been made even in the context of language models prior to LMs. I don't see why this wouldn't have hold for LLMs? Similarly, the observation regarding higher accuracies of LLM predictions on  nodes with higher local homophilic ratio is also a common observation in the graph-mining literature. Finally, in the third insight of this paper, there is no concrete evidence on whether data-leakage could potentially boost capabilities LLMs in leveraging structured information. The lack of evidence is only applicable to prompt styles explored in this paper, but there is no evidence to make any general claims (as also recognized by the authors).

2. The scope of the study is limited to the task of node classification. What about leveraging graph-based information in LLMs in other node tasks such as forecasting or regression problems? Similarly, how could graph based information help LLMs in link prediction problems and its applications in recommendation systems (e.g. predicting if a user will like a movie)?

Some other specific comment:
1. Page 4 line 174: It seems like there is another strategy called k-hop attention strategy which is not described here, but somehow being reffered as "second strategy" in the next paragraph.

**Questions:**

None.

---

> ### Author Response · Authors · 2023-11-18
> **Response to Reviewer Ayoh (1/2)**
>
> Thank you for your thoughtful review. We appreciate your acknowledgment of our paper's detailed experimental study, which delves into the impact of structural information on LLMs in node classification tasks.
>
> We address your concerns in detail below.
>
>
> > W1.1: “Lack of significant contributions. My biggest concern about this paper is that the conclusions derived from the paper are quite trivial and unsurprising. For instance, the observation that stronger improvements in the performance on data with scarce textual features have been made even in the context of language models prior to LMs. I don't see why this wouldn't have hold for LLMs? Similarly, the observation regarding higher accuracies of LLM predictions on nodes with higher local homophilic ratio is also a common observation in the graph-mining literature.”
>
>
> Our intention is to investigate if LLMs can be augmented with appropriate structural information for text classification tasks, which is not reiterating known concepts from graph-mining.
>
> Some other interesting/non-trivial observations:
> 1. **Paper with high homophily is easier for LLMs to classify even without any structural information** (Page 9, Table 5, row 2). With zero-shot prompts (only the title of the paper is given), we observe a significant positive correlation between local homophily ratio and prediction correctness on 4 datasets. This is actually very interesting.
> 2. **The benefit of structural information saturates earlier on LLMs than MPNNs.** To show this, we add an experiment to compare the performance increase from incorporating structural information for LLMs and MPNNs respectively. The results are shown in below table and included in Appendix D.2, Page 16, Table 9.
>
> The average increase from structural data of **LLMs** on 4 datasets is 2.78% (rich context) and 5.44% (scarce context). But the increase from structural data of **MPNNs** is 6.98% (rich context) and 14.07% (scarce context), which is significantly higher than the gain of LLMs. It means that The benefit of structural information saturates earlier on LLMs than MPNNs. This may be explained by LLMs have potential advantages in handling rich text features compared to MPNNs.
>
>
> Table 9: Classification accuracy for the OGBN-ARXIV, CORA, ARXIV-2023, PUBMED on ChatGPT as well as GCN, SAGE and MLP. ↑ (LLMs) denotes the improvements of best prompt style that leverages structural information over zero-shot method. ↑ (MPNNs) denotes the improvements of the best MPNNs over MLP (without structural information).
> | Textual Context | Prompt Style               | Arxiv  | Cora   | Arxiv 2023 | Pubmed |
> |-----------------|----------------------------|--------|--------|------------|--------|
> | Rich            | Zero-shot                  | 74.0   | 66.1   | 73.5       | 88.6   |
> |                 | 1-hop title+label          | 75.1   | 72.5   | 73.8       | 89.1   |
> |                 | 2-hop title+label          | 74.5   | 74.7   | 73.2       | 89.7   |
> |                 | 1-hop attention            | 74.7   | 72.5   | 73.6       | 88.8   |
> |                 | $\uparrow$ (LLMs)          | 1.1    | 8.6    | 0.3        | 1.1    |
> |-----------------|----------------------------|--------|--------|------------|--------|
> |                 | MLP                        | 69.9   | 65.4   | 69.7       | 86.2   |
> |                 | GCN                        | 75.4   | 83.0   | 70.3       | 88.4   |
> |                 | SAGE                       | 75.0   | 83.2   | 70.9       | 90.0   |
> |                 | $\uparrow$ (MPNNs)         | 5.5    | 17.8   | 1.3        | 3.8    |
> |-----------------|----------------------------|--------|--------|------------|--------|
> | Scarce          | Zero-shot                  | 69.8   | 61.8   | 66.6       | 85.9   |
> |                 | 1-hop title                | 72.3   | 69.6   | 70.7       | 80.8   |
> |                 | 1-hop title+label          | 74.3   | 73.9   | 70.4       | 84.7   |
> |                 | 2-hop title                | 71.3   | 69.9   | 68.9       | 83.5   |
> |                 | 2-hop title+label          | 74.2   | 74.5   | 68.5       | 86.4   |
> |                 | $\uparrow$  (LLMs)         | 4.5    | 12.7   | 4.1        | 0.5    |
> |-----------------|----------------------------|--------|--------|------------|--------|
> |                 | MLP                        | 61.9   | 55.7   | 58.5       | 82.0   |
> |                 | GCN                        | 74.8   | 81.2   | 70.3       | 87.1   |
> |                 | SAGE                       | 74.4   | 78.8   | 69.1       | 87.9   |
> |                 | $\uparrow$ (MPNNs)         | 13.0   | 25.6   | 11.8       | 6.0    |

---

> > ### Author Response · Authors · 2023-11-18
> > **Response to Reviewer Ayoh (2/2)**
> >
> > > W1.2 “The lack of evidence is only applicable to prompt styles explored in this paper, but there is no evidence to make any general claims (as also recognized by the authors).”
> >
> > See common response.
> >
> > > W2: “The scope of the study is limited to the task of node classification. What about leveraging graph-based information in LLMs in other node tasks such as forecasting or regression problems? Similarly, how could graph based information help LLMs in link prediction problems and its applications in recommendation systems (e.g. predicting if a user will like a movie)?”
> >
> > See common response.
> >
> > > “Page 4 line 174: It seems like there is another strategy called k-hop attention strategy which is not described here, but somehow being referred to as "second strategy" in the next paragraph.”
> >
> > Thank you for your attention to detail. We actually name the second one as “k-hop attention” in L 182. We introduce two prompting strategies in the paper. The first strategy of prompting is to incorporate randomly selected neighbors (L 168) and the second strategy of prompting is to weigh the influence of neighboring nodes based on attention (L 177).

---

> > > ### Comment · Reviewer_Ayoh · 2023-11-20
> > > **Response to Author's comments**
> > >
> > > I thank authors for their responses and providing more details. However, I am afraid  that I am still not convinced regarding the significance of the contributions. The major key message of this paper is that  LLMs can benefit more from structural information when textual node features are scarce. The benefit is further enhanced in case of homophilic nodes. These both observations are expected to be seen for any textual-feature based models and I don't see why LLMs would have behaved differently. More generally, If you provide more information to a model, it is certainly going to boost the performance. The boost in the performance is expected to be more for less complex models and less for more complex models like LLMs. As of now I am inclined to keep my rating as it is as of now.
> > >
> > > I will read author's responses to other reviewers before making my final decision.

---

> > > > ### Author Response · Authors · 2023-11-21
> > > > **Response to Reviewer Ayoh**
> > > >
> > > > Thank you for your insightful feedback. We'd like to highlight two points indicating that our results are non-trivial.
> > > >
> > > > 1. While it’s true that structural information is generally more helpful when text is scarce, **quantitatively ChatGPT behaves differently from GNNs**: the benefit of structural information saturates much earlier than GNNs with moderate rich textual features; and this is **non-trivial** since LLaMA-2 didn’t saturate as early as ChatGPT. The **average increase** from structural data on 4 datasets for ChatGPT/MPNNs/LLaMA-2-7B-chat are 2.78%/6.98%/21.7% respectively. This quantitative insight would also be instrumental for future studies as LLMs grow more and more powerful. We update our paper to make this point more clear (L 565-569).
> > > >
> > > > 2. The story of homophily is actually quite nuanced: in Table 5, zero-shot prediction (the one that doesn’t use structural information at all) also showed high correlation with the homophily ratio of the node. This suggests a **complicated mechanism for LLMs to perform better on homophilous nodes**: those nodes are easier to be classified in the first place; the added structural information has some further contributions.
> > > >
> > > >
> > > > Overall, these findings are not clear to the community a priori.

---

### Official Review · Reviewer_PJPi · 2023-11-01

**Soundness:** 2 fair
**Presentation:** 3 good
**Contribution:** 2 fair
**Rating:** 3
**Confidence:** 4

**Summary:**

This paper studies the problem of enabling Large Language Models (LLMs) to process graph-structured data. Specifically, this paper studies the potential effects on node classification accuracy with different prompting methods. The paper also studies the effect of data leakage and homophily.

**Strengths:**

S1. This paper is easy to follow.

S2. The proposed methods work reasonably well.

S3. The topic combining Graph and LLM is interesting.

**Weaknesses:**

W1. The review feels the motivation of this paper is not clear. Although LLM is popular, why we need LLMs for node classification tasks is not well-explained. The authors only focus on one node classification task. Given that a simple GNN can already yield satisfactory results for this task, the rationale for introducing LLM, which may be slower in inference, is unclear.

W2. This paper lacks solid explanations and analysis of the experimental results. For example, why do different prompts result in different performance? Is there any theoretical analysis or intuitions? Why can LLM outperform GNN on ARXIV-2023 while not on OGBN-ARXIV? The current version of this paper only provides an experiment report without solid analysis.

W3. The experimental analysis is also weak. The experiments in section 3.4 cannot really show the impact of homophily on LLM's classification accuracy. It may show that incorporating the context information with the same label can improve performance. The authors should compare more ablations with context samples which have the same label.

W4. The authors study the problem of incorporating structural information (such as graphs) to improve the predictive accuracy of LLMs but only conduct experiments in node classification. Some experimental results might provide better understanding. For example, checking the performance of other competitive LLM methods on graphs and downstream tasks.

**Questions:**

Please see the above.

---

> ### Author Response · Authors · 2023-11-18
> **Response to Reviewer PJPi**
>
> Thank you for your thoughtful feedback! We appreciate your recognition of the paper's clarity and the novelty of combining graphs and LLMs. We address your concerns in detail below.
>
> > W1: “The review feels the motivation of this paper is not clear. Although LLM is popular, why we need LLMs for node classification tasks is not well-explained. The authors only focus on one node classification task. Given that a simple GNN can already yield satisfactory results for this task, the rationale for introducing LLM, which may be slower in inference, is unclear.”
>
> Our intention is to investigate if **LLMs**, known for their versatility and robustness in text processing, **can be augmented with appropriate structural information**, instead of **simply applying LLMs to solve graph tasks**. Our motivation can be summarized as follows:
> 1. LLMs can serve as a zero-shot or few-shot classifier without any training, compared to GNNs, which is advantageous in some real-life scenarios such as online recommendations.
> 2. While GNNs are efficient for graph-structured data, LLMs offer potential advantages in handling rich text features within nodes, which might not be fully leveraged by GNNs.
> 3. The conditions under which LLMs can benefit from structural information is underexplored in current literature.
>
> We refined the introduction and discussion sections to better clarify this motivation (L5, L47-48, L135, L142).
>
> > W2.1: “This paper lacks solid explanations and analysis of the experimental results. For example, why do different prompts result in different performance? Is there any theoretical analysis or intuitions?”
>
> It may be explained by that different prompts may encapsulate varying degrees of contextual relevance to the target node, impacting the LLM's ability to utilize the provided information effectively. This is further discussed in Section 3.4: Impact of Homophily on LLMs Classification Accuracy.
>
> > W2.2: "Why can LLM outperform GNN on ARXIV-2023 while not on OGBN-ARXIV?"
>
> The relationship of performance of GNNs and LLMs are dependent on datasets and prompts. The observation that "LLM outperforms GNN on ARXIV-2023 while not on OGBN-ARXIV" may be explained by that LLMs are better at classifying certain types of papers. And the two datasets have different label distribution.
>
> > W3. “The experimental analysis is also weak. The experiments in section 3.4 cannot really show the impact of homophily on LLM's classification accuracy. It may show that incorporating the context information with the same label can improve performance. The authors should compare more ablations with context samples which have the same label.”
>
> In Section 3.4, two key experiments were conducted to investigate the impact of homophily on LLM classification accuracy.
> 1. The first experiment involved dropping neighbors of a target node in three ways: those with the same label, different labels, and randomly. This demonstrated that a higher local homophily ratio, where more neighbors share the same label as the target node, positively influences LLM's prediction accuracy.
> 2. The second correlation study **revealed a positive correlation** between **local homophily ratio** and **prediction correctness in all 5 datasets** with p value less than 0.001 (Page 9, Table 6). **Notice this experiment does not involve any dropping of neighbors and is calculated in the original dataset**. This further emphasizes the significance of homophily in improving the accuracy of LLM classifications.
>
> While we believe the results of the two experiments are fairly strong evidence of the impact of homophily, we still appreciate it if you could elaborate on the details of the ablation setting suggested in your review, so that we can further improve our experiments. Thanks!
>
> > W4. “The authors study the problem of incorporating structural information (such as graphs) to improve the predictive accuracy of LLMs but only conduct experiments in node classification. Some experimental results might provide better understanding. For example, checking the performance of other competitive LLM methods on graphs and downstream tasks.”
>
> Thank you for your advice on checking “other competitive LLM methods on graphs and downstream tasks”. We acknowledge that other graph-related problems are important to understand how LLMs can benefit from structural information. However, as an early study on this topic, we aim to focus more on the depth of the analysis (answering the why question) rather than the breadth. We believe narrowing down the type of prediction tasks could provide us a better testbed for our analysis. That being said, we plan to extend our research to encompass a broader range of graph-related tasks, including link prediction, in future work.

---

> > ### Author Response · Authors · 2023-11-21
> > **Response to Reviewer PJPi**
> >
> > Dear Reviewer PJPi, thank you again for your effort in reviewing our paper. As the end of the discussion period approaches, we would like to ask if our responses were able to sufficiently address your concerns. If you have further questions, please let us know and we are eager to further address them!

---

### Official Review · Reviewer_ytsy · 2023-11-02

**Soundness:** 3 good
**Presentation:** 4 excellent
**Contribution:** 3 good
**Rating:** 6
**Confidence:** 3

**Summary:**

This paper investigates how Large Language Models (LLMs) can leverage graph structural information for node classification tasks with textual features.  The authors investigate the "when" and "why" of incorporating structured data, particularly graphs, into LLMs. They examine different prompting methods and factors that affect the performance of LLMs, such as data leakage, homophily (nodes with similar characteristics). Through controlled experiments and correlational analyses, the authors  establish a positive relationship between the local homophily ratio and the prediction accuracy of LLMs, i.e., more homophily incorporated in the prompt, the more accurate the prediction becomes. While data leakage is a potential concern , the authors observe that LLMs' consistent performance across different datasets suggests that they are robust and can perform well across varying distribution domains. Finally, the authors provide insights and suggestions for future research on LLMs and graph data.

**Strengths:**

* The paper is well-written, clear, and organized. The authors present their motivation, approach, methodology, results, and contributions in a logical and coherent manner.
* The paper addresses an interesting topic of integrating LLMs with graph data, which is a crucial data modality that can provide additional information to LLMs.
* The evaluation results provide valuable insights into the potential and limitations of LLMs in leveraging graph structural information, as well as the challenges and opportunities for future research on LLMs and graph data.

**Weaknesses:**

One of the limitations is that the study only focuses on node classification tasks and does not explore other graph-related tasks. It is unclear whether these techniques can be generalized to other tasks.

**Questions:**

Do you think other Large language models such as LLaMA or Falcon robust against data leakage?

---

> ### Author Response · Authors · 2023-11-18
> **Response to Reviewer ytsy**
>
> Thank you for recognizing the “valuable insights” in our work and clear writing. We address your detailed concerns below.
>
>
> > One of the limitations is that the study only focuses on node classification tasks and does not explore other graph-related tasks. It is unclear whether these techniques can be generalized to other tasks.
>
> See common response.
>
> > Question: "Do you think other Large language models such as LLaMA or Falcon are robust against data leakage?"
>
> Similar to ChatGPT, There is no significant evidence that the performance of LLaMA-2 is significantly attributed to data leakage.
>
> We run the classification experiment on LLaMA-2-7B-chat [1] and include the results below as well as in Appendix D, Page 16, Table 8. Notice that the pretraining data of  LLaMA-2 has a cutoff of September 2022 (Page 77) and the test data in arxiv-2023 are arXiv papers published in 2023.
>
> If LLaMA-2-7B-chat rely on data leakage to make accurate prediction, the performance drop of LLaMA-2-7B-chat between ogbn-arxiv (may have leakage problem) and arxiv-2023 (leakage-free) should **be significantly greater** than the drop on MPNNs on two datasets. This is because LLMs may benefit from their memory on ogbn-arxiv from data leakage to have higher accuracy. But this advantage is not likely on arxiv-2023. However, in Table 8 (below), we **do not observe this**: the performance drop of LLMs between ogbn-arxiv and arxiv-2023 **is less than** the drop on MPNNs in scarce context (6.7% vs 4.5%), and slightly higher in rich context (4.6% vs 5.1%). From this, we argue that there is no significant evidence that the performance of LLaMA-2 is significantly attributed to data leakage.
>
>
> | Textual Context | Prompt Style       | Ogbnarxiv | Arxiv |
> |-----------------|--------------------|-----------|-------|
> | Rich            | Zero-shot          | 45.1      | 45.1  |
> |                 | 1-hop title        | 51.6      | 50.0  |
> |                 | 1-hop title+label  | **66.9**  | **60.2**|
> |                 | GCN                | 75.4      | 70.3  |
> |                 | SAGE               | 75.0      | 70.9  |
> | Scarce          | Zero-shot          | 38.8      | 38.2  |
> |                 | 1-hop title        | 51.5      | 45.5  |
> |                 | 1-hop title+label* | **58.0**  | **53.4** |
> |                 | GCN                | 74.8      | 70.3  |
> |                 | SAGE               | 74.4      | 69.1  |
>
>
>
>
> [1] Llama 2: Open Foundation and Fine-Tuned Chat Models, https://arxiv.org/abs/2307.09288

---

> > ### Author Response · Authors · 2023-11-21
> > **Response to Reviewer ytsy**
> >
> > Dear Reviewer ytsy, thank you again for your effort in reviewing our paper. As the end of the discussion period approaches, we would like to ask if our responses were able to sufficiently address your concerns. If you have further questions, please let us know and we are eager to further address them!

---

> > > ### Comment · Reviewer_ytsy · 2023-11-23
> > >
> > > Dear Authors, thank you for your detailed rebuttal and additional experiments. I appreciate your effort to clarify some concerns raised by other reviewers. I think you've provided some evidence that LLaMA-2 isn't much affected by data leakage, and that your method can perform well on node classification tasks with textual features. I still have some doubts about how your approach can generalize well for other graph-related tasks and other datasets, therefore I keep my recommendation as marginally Accepted.

---

### Official Review · Reviewer_2d38 · 2023-11-03

**Soundness:** 3 good
**Presentation:** 3 good
**Contribution:** 3 good
**Rating:** 6
**Confidence:** 3

**Summary:**

This paper presents an empirical study on how LLMs can capture information contained in graph structures. Specifically, the authors consider the task of node classification with textual features as node features. The authors verify that with neighbor information to complement textual features, the performances of LLMs can be improved, especially under the scenario where scarce textural information is given. The authors then analyze where the LLMs performance on graphs come from, and come to two conclusions: the performance of LLMs on graphs is not related to data leakage, and homophily contributes to the performance of LLMs on graph-structured data.

**Strengths:**

1. Timely topic. It is interesting to explore the potentials of LLM on a wide range of tasks, and graph node classification task with rich textual information is a suitable topic for LLMs, in that it is closely related to texts and incorporates different data structures.
2. Good organization. The paper is well-organized and easy to follow. It is easy for me to understand the main ideas and conclusions/arguments of this paper.
3. The 'data leakage' aspect of this paper is interesting and a somewhat unique aspect in the combination of LLMs and graphs. Indeed, in the area of traditional graph learning, data leakage is not an issue. Therefore, I appreciate the authors' efforts in identifying such a potential cause and perform experiments.

**Weaknesses:**

1. The conclusions of this paper are somewhat not surprising. As LLMs work on texts, it is intuitive that with rich texts, LLMs can already well understand the node, and when there are scarce texts, some additional related texts can further boost the performance of LLM. Also, the conclusion about homophily is also not surprising: bringing heterophilous nodes to the context will distract LLM and thus compromise the accuracy. I am not saying that the contribution of this paper is limited --- **it is a solid paper with good contributions**, but the conclusions are just not that surprising, which would slightly reduce my rating.
2. Studying the behavior of ChatGPT indeed leads to better understanding of LLMs on graphs. However, as ChatGPT is not open source, it would be better if the authors can also perform experiments on open-source models, such as LLaMA. In that way, the insights may better lead to more concrete efforts in combining LLMs with graph learning, as LLaMA can be more easily fine-tuned to fit the need of graph data understanding. Also, as LLMs are sensitive to prompts,  there is no guarantee that the conclusions on ChatGPT can generalize to other LLMs. In this sense, it is always better to do experiments on a wider range of LLMs to ensure that the conclusion is general enough.

**Questions:**

1. Do you observe cases where ChatGPT fails to follow the instructions and generates stuff that cannot be automatically parsed? How do you deal with these samples?
2. LLMs may be sensitive to the order of contexts. Have you tried to modify the order of 1-hop, 2-hop neighbors in the 1/2-hop title+label setting? Does it lead to significant performance changes?

**Details Of Ethics Concerns:**

Not required.

---

> ### Author Response · Authors · 2023-11-18
> **Response to Reviewer 2d38**
>
> Thank you for recognizing the timeliness and good organization of our work, and the novel
> aspect of data leakage. We address your detailed concerns below.
>
> > it would be better if the authors can also perform experiments on open-source models, such as LLaMA.
>
> See common response. We added experiments on LLaMA-2-7B-chat [1].
>
> > Question1: "Do you observe cases where ChatGPT fails to follow the instructions and generates stuff that cannot be automatically parsed? How do you deal with these samples?"
>
> We did not observe that ChatGPT failed to follow the instructions. For example on dataset ogbn-arxiv, our instruction is "Please predict the most appropriate arXiv Computer Science (CS) sub-category for the paper. The predicted sub-category should be in the format ’cs.XX’.". ChatGPT always prints out 'cs.XX' as its answer without extra text.
> On the other hand, LLaMA-2-7B-chat sometimes provides extra explanation text in addition to its prediction. In this case, we matched the last appeared category as its prediction. This parsing strategy worked throughout our experiment.
>
> > Question 2: "LLMs may be sensitive to the order of contexts. Have you tried to modify the order of 1-hop, 2-hop neighbors in the 1/2-hop title+label setting? Does it lead to significant performance changes?"
>
> Shuffling the order of neighbors in prompt has a negligible effect on prediction performance. We have conducted the shuffling experiment on Cora, and the results show small standard deviations after shuffling the order of neighbors over 3 runs: $0.7281 \pm 0.0031$, (1-hop title+label), $0.6359 \pm 0.0008$ (1-hop title), $0.7577 \pm 0.0031$ (2-hop title+label). This observation also holds for other datasets in our experiments.
>
>
> [1] Llama 2: Open Foundation and Fine-Tuned Chat Models, https://arxiv.org/abs/2307.09288

---

> > ### Comment · Reviewer_2d38 · 2023-11-20
> > **Response acknowledged.**
> >
> > I have read the author response. I appreciate the authors providing new results on LLaMA. I will maintain a weak accept for now and will consider raise the scores in future reviewer discussions.

---

### Author Response · Authors · 2023-11-18
**Common response to all reviewers (1/2)**

We thank all reviewers for their insightful and constructive comments on our work. The timeliness and organization of our paper were commended by **Reviewer 2d38** and **Reviewer PJPi**. The novel approach towards 'data leakage' within LLM and graph data, highlighted by **Reviewer 2d38**, was seen as both “interesting” and “unique”. **Reviewer ytsy** and **Reviewer Ayoh** praised the clear, organized, and detailed nature of our experimental study, affirming the soundness of our methodology. Lastly, the insights into the potential and limitations of LLMs provided by our evaluation is recognized by **Reviewer ytsy** and **Reviewer vRFw**.



**Recap: What is our goal?** Investigate when and why LLMs can be augmented from additional structural information on the task of classifying texts. “Classifying text with more text”.

**Recap: What is not our goal?** Develop prompts to solve node classification problems.



Here we want to make some general comments to help clarify the purpose of this paper.


> @Reviewer ytsy, PJPi, Ayoh: This study is limited to node classification tasks.

While there is a natural map between our work and node classification, our major focus is to examine **how LLMs can benefit from additional structural information**. Document classification is a very classic language task, and we found it can be **naturally augmented with structural context** by borrowing popular node classification datasets. This extension is ubiquitous. For example, some tweets may come with some intrinsic relationship with other tweets (e.g. repost), then it is interesting to ask whether LLMs can leverage these extra tweets to improve the classification of the target tweet.

In comparison, it is not very well-motivated to perform link prediction tasks in our study as there is not a corresponding language task. Though we still believet the proposed method can be extended to broader tasks like link prediction with meaningful results.

We made a few modifications in our paper to make the purpose of this paper more clear, highlighted with the red font.


> @Reviewer 2d38, ytsy: More LLMs are needed.

We have included a table below for LLaMA-2-7B-chat [1]. The result is also included and discussed in Appendix D, Page 16, Table 8. The results align with our observation in the paper with ChatGPT that incorporating structural information actually brings performance improvement in both rich and scarce contexts. But a different observation is that the improvement in scarce textual context is not necessarily higher than the improvement in rich textual context. This may be because LLaMA-2 is not able to sufficiently leverage the entire text for the prediction in zero-shot prediction.

Combining the results of ChatGPT, the conclusion is that, with powerful enough LLM and rich text (e.g. ChatGPT with rich context), the structural information is marginal. But when the text information is scarce or if the LLM cannot fully utilize the text information, structural information can be significantly helpful.

| Textual Context | Prompt Style      | Ogbnarxiv | Cora     | Arxiv    | Pubmed   | Ogbnproduct |
| --------------- | ----------------- | --------- | -------- | -------- | -------- | ----------- |
| Rich            | Zero-shot         | 45.1      | 18.1     | 45.1     | 71.6     | 51.3        |
|                 | 1-hop title       | 51.6      | 51.5     | 50.0     | 68.8     | 52.1        |
|                 | 1-hop title+label | **66.9**  | **66.7** | **60.2** | **73.0** | **77.2**    |
|                 | $\uparrow$        | 21.8      | 48.6     | 15.1     | 1.4      | 25.9        |
| Scarce          | Zero-shot         | 38.8      | 24.5     | 38.2     | 70.1     | 51.7        |
|                 | 1-hop title       | 51.5      | 44.8     | 45.5     | 70.9     | 52.8        |
|                 | 1-hop title+label | **58.0**  | **71.0** | **53.4** | **75.5** | **78.9**    |
|                 | $\uparrow$        | 19.2      | 46.5     | 15.2     | 5.4      | 27.2        |

---

> ### Author Response · Authors · 2023-11-18
> **Common response to all reviewers (2/2)**
>
> > @Reviewer Ayoh, vRFw: The logic behind the data leakage experiment is not clear/may not generalize.
>
> 1. We tried quite a few different prompt styles including zero-shot, few-shot, Chain-of-Thought, 1/2-hop title(+label) and 1-hop attention prompts (Table 4). The conclusion of data leakage based on these prompts may generalize to other prompting styles.
>
> 2. We did have concrete evidence of the minor effect of potential data leakage for the prompting styles that we tried. If LLMs rely on data leakage to make accurate prediction, the performance drop of LLMs between OGBN-ARXIV (may have leakage problem) and ARXIV-2023 (leakage-free) should **be significantly greater** than the drop on MPNNs on two datasets. This is because LLMs may benefit from their memory on OGBN-ARXIV from data leakage to have higher accuracy. But this advantage is not likely on ARXIV-2023. However, in Table 4, we **observe exactly the contrary**: the performance drop of LLMs between OGBN-ARXIV and ARXIV-2023 **is less than** the drop on MPNNs on two datasets (1.3% compared to 5.1% in rich context, 3.6% vs 4.5% in scarce context). This means that LLMs actually generalize well to leakage-free data. We have revised the paper to make this argument more clear (L 256-262).
>
>
> [1] Llama 2: Open Foundation and Fine-Tuned Chat Models, https://arxiv.org/abs/2307.09288